# High-Throughput Bit-Pattern Matching under Heavy Interference on FPGA

**Dimitris Nikolaidis [1,*], Panos Groumas [2] , Christos Kouloumentas [2] and Hercules Avramopoulos [1]**

[1] School of Electrical and Computer Engineering, National Technical University of Athens, 15773 Athens, Greece
[2] Optagon Photonics, Eleftheriou Venizelou 47, 15351 Pallini, Greece
* Correspondence: dimnikolaidis@mail.ntua.gr

**Abstract:** Bit-pattern matching is an important technological capability, used in many fields such as network intrusion detection (NID) and packet classification systems. Essentially, it involves the matching of an input bit pattern to a bit-pattern entry of a memory structure inside the system. Contemporary methods focus on the decomposition of the input bit pattern into smaller and more manageable parts, with the subsequent parallel processing of these elements. This fragmentation promotes the use of advanced pipeline techniques and hardware optimizations, enabling these methods to achieve very high throughputs and reasonable efficiency. However, the functionality of their respective circuits is limited to only performing pattern matching when there is no interference. In this article, we intend to present a circuit that performs pattern matching under heavy interference; instead of fragmentation, a more holistic approach will be adopted. To improve the throughput of the circuit, long bit sequences will be directly compared to many memory entries simultaneously. The minimization of hardware consumption and maximization of efficiency in these comparisons will be achieved with the use of novel hardware architecture that is based on pipelined adder trees and comparators. The platform of implementation is an FPGA (Field-Programmable Gate Array).

**Keywords:** bit-pattern matching; high interference; high throughput; VHDL; FPGA

## 1. Introduction

Bit-vector pattern matching is an important function of modern hardware systems and has a strong presence in many fields. Notable examples include packet header classification [1], string matching (in the field of network intrusion systems) [2–4], and many others, including biology, image processing, and text search [5]. In essence, bit-vector pattern matching concerns the identification of an input pattern of a specific size to a known pattern that resides in the memory of the system performing the operation [6]. Since the size of the patterns is typically greater than 1000 bits, the main method used by state-of-the-art solutions is to break down the input pattern into smaller sub-patterns and then use small and efficient processing elements to perform pattern matching at maximum throughput [6]. In this paper, we will present a different approach to pattern matching that focuses on matching under heavy interference.

Contemporary methods are very efficient in pattern matching; however, there is a critical disadvantage connected to their use, in that they cannot perform pattern matching when the pattern is significantly altered. While there are many solutions for pattern matching that are very proficient [6,7] and employ advanced methods to maximize operating frequency and the optimal use of memory blocks, there is little to no provision for performing pattern matching at the same data rates (gigabits) under interference. Since digital data can be manipulated easily and can take any form, being able to distinguish between very large patterns that have undergone significant alternation can prove to be very important in fields such as image recognition [8].

The circuit presented in this article aims to solve this problem by changing the philosophy behind the process of pattern matching. Instead of breaking down the pattern into smaller sub-patterns, the entire pattern is directly compared with as many memory elements as possible in every clock cycle. To achieve this end with few resources, the comparison procedure was augmented with pipelined adder trees. The intention behind this choice is to sever the connection between the size of the pattern and the operating frequency. In other words, the circuit is able to handle large patterns without suffering losses in operating frequency and throughput. The architecture used is minimalistic and can be implemented very easily on all types of hardware platforms. The circuit can handle bit-error rates (BER) of up to $10^{-1}$ with high accuracy, as a result of its nature. Detection via adder trees and the comparators of large bit patterns is resistant to bit errors, regardless of the error pattern.

This article is, first and foremost, an introductory presentation of the novel treatment of pattern matching that is described above. It is not meant to be a conclusive survey of every single possible configuration of the architecture of the proposed circuit but is, rather, a proof of concept. It presents evidence that there is, in fact, merit in pursuing this type of approach to pattern matching since it displays multiple advantages that are not present in other schemes. The circuit itself is a kernel implementation. Certain aspects of it (e.g., memory) were implemented without particular attention being paid to their efficiency, as a means to an end, and other aspects (I/O circuitry) are left to the user. Since these functions are already widely researched, future plans include the integration of optimized memory and an I/O circuitry module, as well as a thorough survey of all the parameters connected to the design.

This article consists of 6 sections. The second section houses the presentation of a set of fundamentals regarding the function of pattern matching itself, as well as the philosophy behind using adder trees instead of only comparators. This will become the basis of our approach. The third section presents the fundamental pattern-matching circuit upon which the general design is based. Its function is to compare the entire pattern to a memory entry and conclude whether it matches the pattern being searched. The fourth section covers the functionality of the entire circuit, together with the results of its implementation on the chosen board. The fifth section is entirely dedicated to the testing of the pattern-matching mechanism under heavy interference. The main topics are the error model used to simulate the communication channels and the circuit's matching capabilities. The article concludes with a summary of our findings.

## 2. Fundamentals

The main purpose of bit-vector pattern matching is the matching of an input pattern to one known pattern. Usually, the known patterns are referred to as rules and the size of the set of rules is known as the rule set. Rule sets vary in size; generally, they are lower than or equal to 1000 [7]. In other implementations, the patterns are broken down further into sub-patterns, but this will not be the case in our circuit. Table 1 presents an input pattern, along with the rule set used. The rule size is 5 and the length of the pattern is 10.

**Table 1.** Rule set.

| Input Pattern | 0011011101 |
|:---:|:---:|
| Rule 1 | 1101001101 |
| Rule 2 | 0001011101 |
| Rule 3 | 1001011111 |
| Rule 4 | 1111110101 |
| Rule 5 | 1011110100 |

In this particular case, the input pattern does not match any entry exactly, but it can be matched with Rule 2 since it is the closest in terms of Hamming distance. Determining the closest bit vector in terms of distance can be simply achieved by xnoring(the bits in

respective positions pass through xnor gates) the input pattern with each memory entry and then summing the product bits. Table 2 depicts this operation.

**Table 2.** The similarity of patterns.

| Input Pattern | Rule 2 | Xnor | Sum of Xnor Gates |
|---|---|---|---|
| 0 | 0 | 1 | |
| 0 | 0 | 1 | |
| 1 | 0 | 0 | |
| 1 | 1 | 1 | |
| 0 | 0 | 1 | 9(1001) |
| 1 | 1 | 1 | |
| 1 | 1 | 1 | |
| 1 | 1 | 1 | |
| 0 | 0 | 1 | |
| 1 | 1 | 1 | |

The highest value for the sum is the number of bits when both patterns are the same and is set at 0 when they are complements. The simplicity of this method fits the minimalistic tone of the circuit. Achieving its implementation with the fewest possible resources and the highest possible frequency, while providing maximum accuracy, will be key to success. Before moving on to the hardware requirements, we are going to present two basic circuits that will be used extensively in the architecture.

*2.1. Adder Tree*

An adder tree is a well-known circuit that calculates the sum of a set of numbers. Its general form can be seen in Figure 1 for a model with 11 bits (each input is a bit).

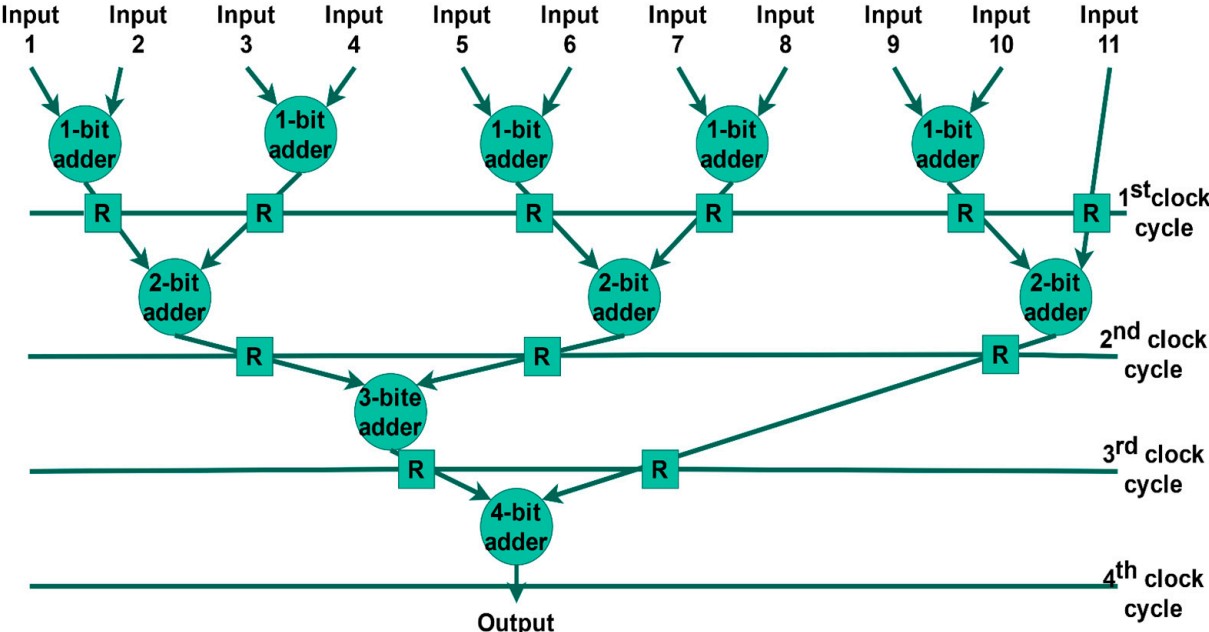

**Figure 1.** Adder tree for an 11-bit model.

The circles are adders of appropriate size for each level, while the squares represent registers. The addition of registers between the levels of adders enables pipelining. At each level, the numbers are added in pairs. If there is a leftover number, it proceeds as previously set. This means that in level $i$, there are $\frac{n}{2^i}$ additions (either ceil $\left(\frac{n}{2^i}\right)$ or floor $\left(\frac{n}{2^i}\right)$) and ceil($\log_2 n$) levels in total. The final sum is also ceil($\log_2 n$) bits in length. The largest adder needed is ceil($\log_2 n$) − 1 bits. The critical path of this adder determines the

maximum operating frequency of the entire tree. This is the main reason the adder tree by itself cannot support very long sequences. Adders of 32 bits are significantly slower than adders of 4 bits when they are forced to perform addition in one clock cycle. Pipelining the bigger adders will result in enormous hardware consumption, which will ultimately offset the trees' efficiency. For the circuit to be able to handle bit sequences of 1000 bits while operating in the frequency of 4-bit adders, with no significant hardware overhead, significant modifications need to be made.

### 2.2. Comparator Tree

A comparator tree is a variation of the adder tree. Just as the name implies, when given a set of $n$ numbers, it can determine the maximum or minimum of these numbers. It can be easily pipelined using the same method as the adder tree. By keeping the position of each number on a parallel register, it can output not only the maximum value but also its position inside the set. The depth of the pipeline is also calculated as ceil($\log_2 n$) clock cycles. In the context of the circuit, this will be used to determine which rule the input sequence is closest to and provide it as an output.

### 3. Core Circuit Description

The circuit performing the matching between the input pattern and one entry of the memory structure will be built in two phases. In the first phase, we are going to present the basic circuit, the lesser pattern identifier upon which the entire architecture will be based; in the second phase we are going to present the scaled-up model, the cascaded pattern identifier, which performs pattern matching with the same operation frequency, regardless of the size of the pattern.

### 3.1. Lesser Pattern Identifier

We assume that the entire input pattern is $n$ bits long. We xnor the input pattern with one memory entry (respective bit positions are fed to xnor gates). Then we break down these bits in $q$ symbols of $m$ bits each, so we have $q \times m = n$. This fragmentation is not aimed at creating sub-patterns. It is a strategy, the ultimate purpose of which is to enable the circuit to perform pattern matching at maximum frequency. Each symbol of the $m$ xnored bits is then driven through a pipelined adder (Section 2.1). The sum of the xnor outputs gives us an estimation of how close the input pattern to our pattern is. After the adder tree, each sum is passed through a comparator. If the sum is equal to the number of bits per symbol ($m$), then the output of the comparator becomes 1, indicating that the correct symbol was detected. Then, by checking the output of the comparators, we can determine how many symbols were detected. If all the $m$ symbols were detected, then the pattern was detected. An implementation of it is shown in Figure 2 for $q = 3$ and $m = 5$. All values are marked on the wires of the circuit for convenience and clarity. The number that the comparators use to determine whether the symbol was detected or not is known as the weight of the lesser identifier. In Figure 2, the weight of the identifier is 5.

It is clear that the lesser pattern identifier alone cannot be used to identify large patterns ($n > 100$). If there are too many symbols ($q >> 1$) then the outputs of the comparators are too many to check in one cycle, thus increasing the critical path. If the symbols are too large, the adder trees require adders comprising many bits, which increases the critical path. For these reasons, the actual pattern matching will be performed using the cascaded pattern identifier.

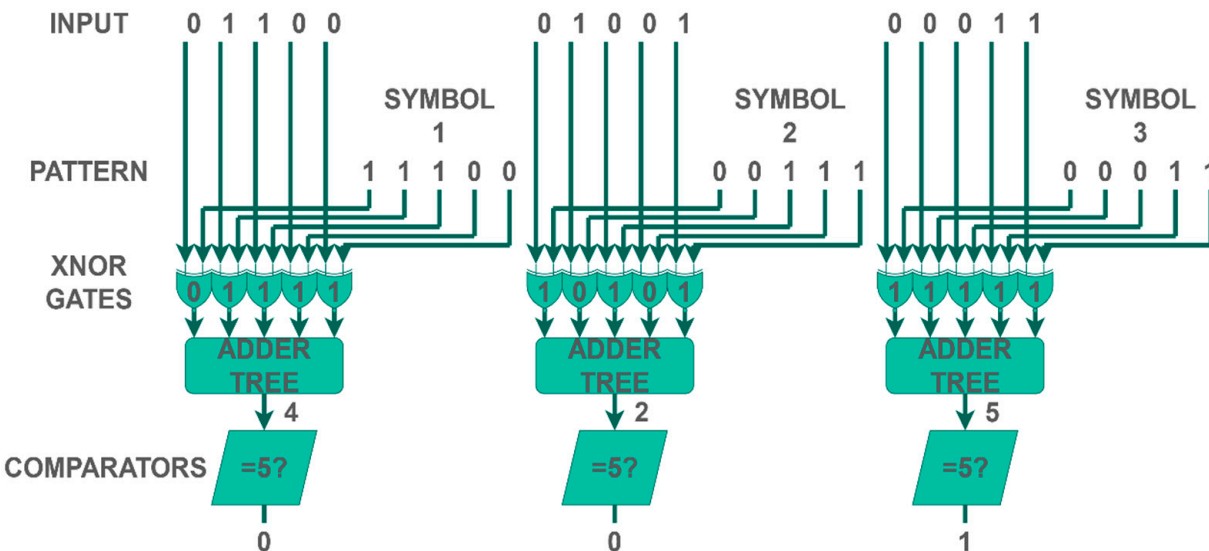

**Figure 2.** The lesser pattern identifier.

*3.2. Cascaded Pattern Identifier*

The cascaded pattern identifier is a circuit with multiple stages. Each stage comprises a lesser pattern identifier (see Section 3.1) where the comparator outputs are fed to the identifier of the next stage. Figure 3 presents the circuit of a cascaded pattern identifier. The number of input bits is kept low in the example, in order to plot the entire circuit clearly. This version has 2 stages. The first stage (stage 1) comprises 30 bits, organized as 5 bits per symbol for 6 symbols. In the second stage, the inputs drop from 30 to 6. They are organized in 2 symbols of 3 bits each. The last 2 bits indicate if the entire pattern of 30 bits was detected. We can see that the weights of the comparators are now higher than or equal to 4 for the first stage and higher than or equal to 2 for the second stage. This means that there is a margin for error. Additionally, the pattern that needs to be detected only appears in the symbols of the first stage. For stage 2, all symbols are replaced with combination 111 since subsequent stages are used to determine how well the first stage detected the pattern. As we can see, only a part of the pattern was detected based on the weights used by the comparators and there is no match. The latency of the circuit is represented by ceil($\log_2 5$) + 1 + ceil($\log_2 3$) + 1 = 7 clock cycles. The term 1 + 1 represents the clock cycles needed by the comparators. To conserve hardware, comparators can be implemented as combinatorial circuits so they do not consume clock cycles. The log terms are a product of the adders. In general, for a circuit of $t$ stages, the clock cycles needed by the circuit are $Cycles = t + \left[\sum_{i=1}^{t} \text{ceil}(\log_2 b_i)\right]$, where $b_i$ represents the bits per symbol for each stage.

By constructing the pattern identifier in this way, we have achieved two important things. Firstly, for large patterns, adders do not need to scale indefinitely. We can use more stages and steadily decrease the number of bits to keep the bits per symbol low. This means that the size of the pattern, in bits, has no effect on the critical path (and as a result of the operating frequency and throughput). Secondly, by manipulating the weights of the comparators, we can adjust the detecting ability of the circuit (it resembles a neural network, where comparators are neurons). As an example, we offer the implemented version of the circuit with 1200 input bits and 3 stages. The general form can be seen in Figure 4. This method can be scaled to any pattern length and any combination of adder-size, and the levels can be constructed and tested to maximize the detection capability.

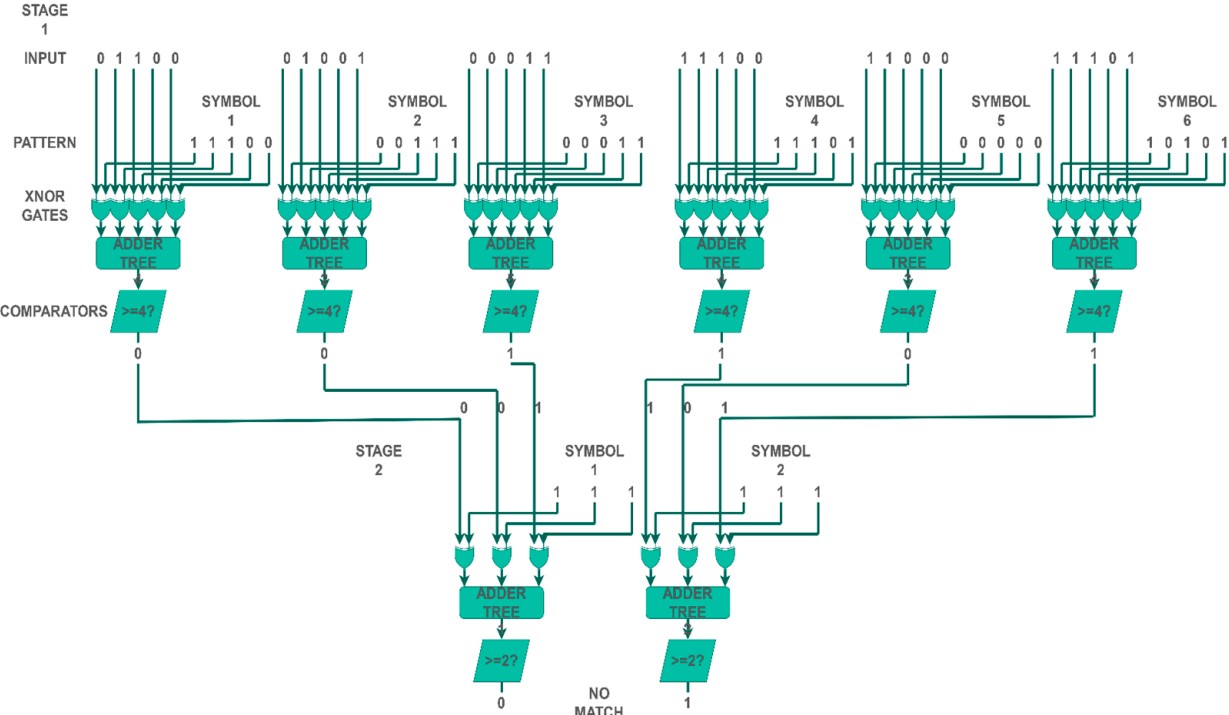

**Figure 3.** The 2-stage cascaded pattern identifier.

In the first stage (lesser identifier 1) the 1200 bits are broken down to 120 symbols of 10 bits. Following the logic of Figure 2, we use 120 adder trees, divided into 10 bits each. The results of these trees are fed to 120 comparators, which have a predetermined weight. The results of the comparators are now fed to the next lesser pattern identifier (2). This identifier has an input of 120 bits, which is broken down into 12 symbols of 10 bits. The same process is applied, then the outputs of the second identifier (14 bits) are fed to the last identifier (3), where they are broken down into 1 symbol of 12 bits. The output of the last identifier is 1 bit. This bit vector is used to determine the detection of a pattern. If the bit output is 1, the pattern was detected. As mentioned, the key pattern of the first lesser identifier is the pattern being searched. For the second and third stages, the key pattern is the all-1 vector. This is because the second- and third-stage identifiers act as confirmation that the pattern in the previous stage was indeed detected. The clock cycles for 1200 bits are $Cycles = 3 + 2 \times \text{ceil}(\log_2 10) + \text{ceil}(\log_2 12) = 15$. The largest adder used is of 4 bits.

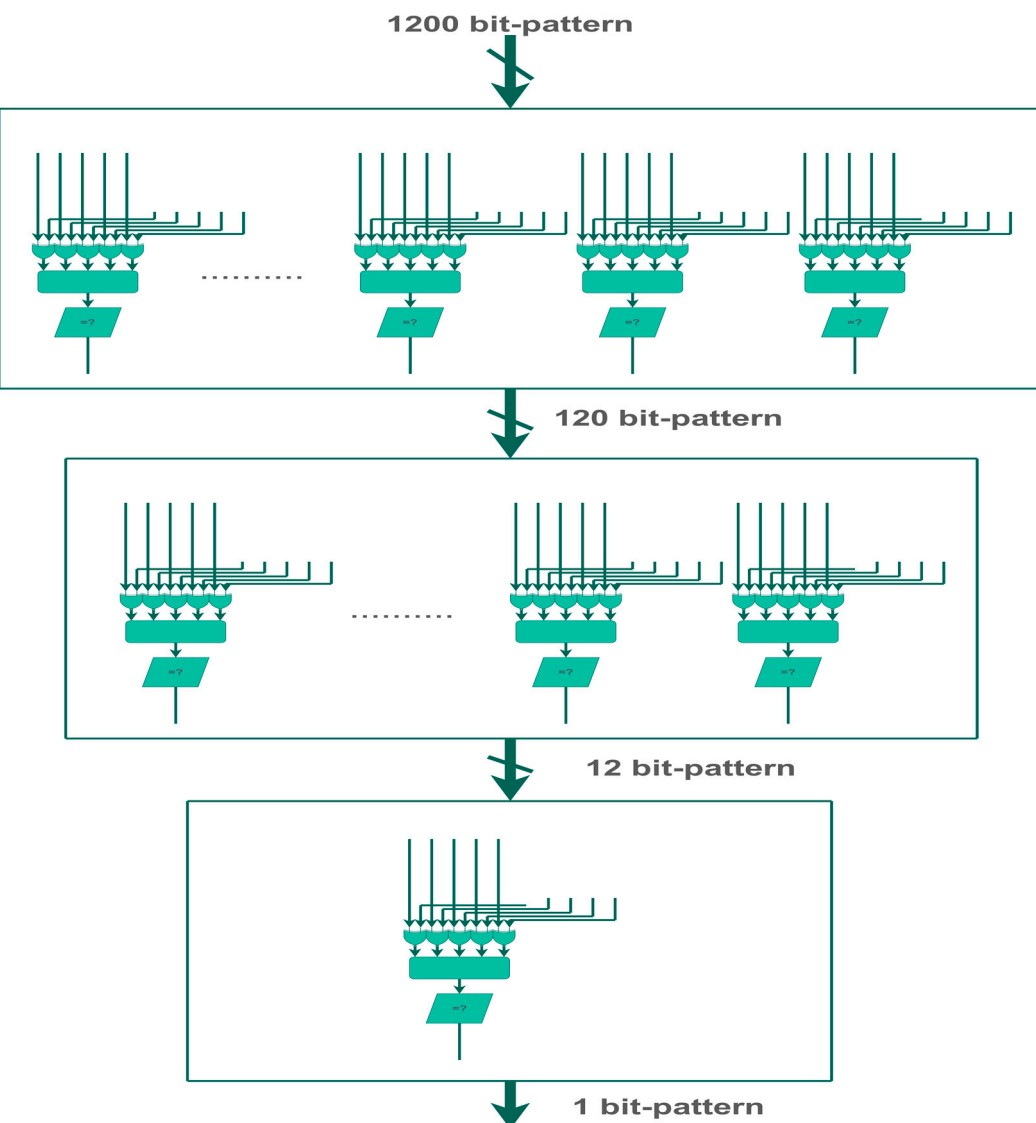

**Figure 4.** The 3-stage cascaded pattern identifier.

## 4. Pattern-Matching Circuit and Implementation

By using the cascaded pattern identifier, we performed pattern matching of an input to a set of memory entries. The number of units of the cascaded pattern identifier is the same as the number of memory entries. The input is fed to all of the units, while each memory entry is fed to its respective unit. If we have enough stages, we can minimize the set of output bits of the cascaded identifier to 1 bit for the final level. If the number of units is $k$, they produce $k$ bits, each bit indicating whether the input pattern was matched correctly. When a pattern is found, these $k$ bits contain one "1" and $k-1$ "0"s. The place of the 1 is the matched entry in the memory. Different approaches can be adopted to detect the 1. In the current case, a comparator tree was used (Section 2.2). The tree's hardware consumption was extremely small and had no detrimental effect on the operating frequency. The general form of their entire pattern identifier can be seen in Figure 5. The rule set consists of 64 entries.

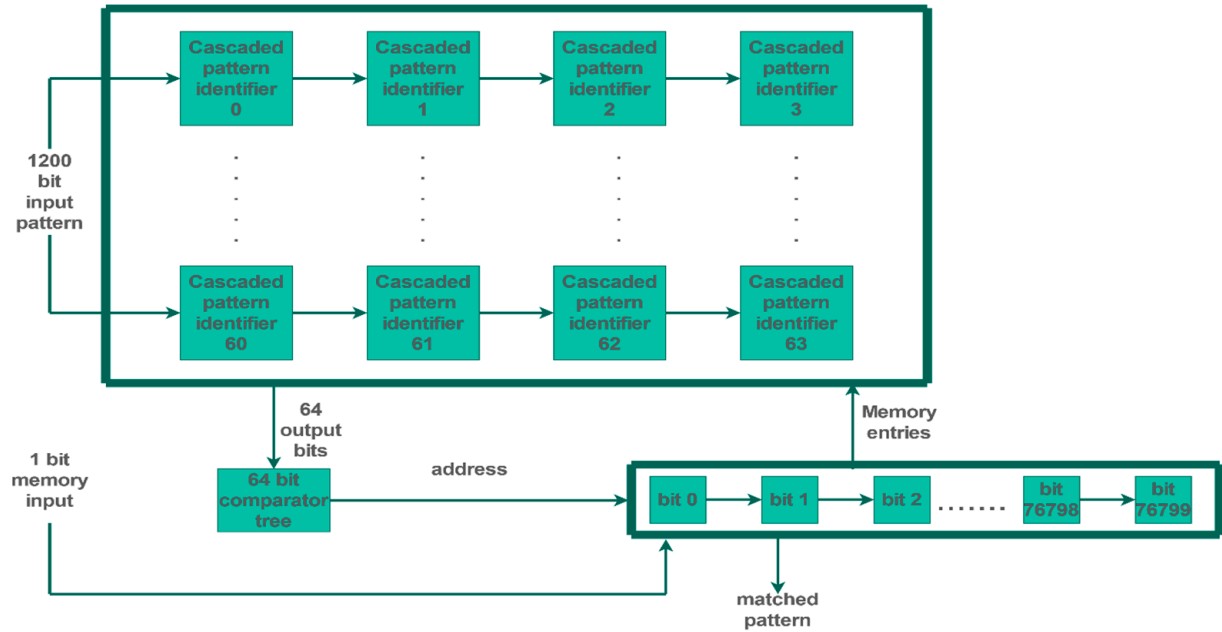

**Figure 5.** Pattern-matching circuit.

The circuit uses the address produced by the comparator tree to output the matched memory entry. The total latency of the circuit is the latency of the cascaded pattern identifier and the latency of the comparator tree at the end, which is 15 + 8 = 23 cycles. In terms of the memory, this was implemented as a very long linear shift register (bits 0 to 1199 are entry 0, bits 1200 to 2399 are entry 1, bits 2400 to 3599 are entry 2, etc., all the way to entry 63), which runs on the same clock as the cascaded identifiers, i.e., 667 MHz) and has a 1-bit input on the first position. If entry $y$ needs to be outputted, the corresponding bits are in positions $y * 1200$ to $(y + 1) * 1200 - 1$. To update the entire memory, $1200 \times 64 = 76{,}800$ clock cycles ($10^{-4}$ s) are needed (1 bit per clock cycle). To update parts of the memory, a proportional number of clock cycles are needed; however, updates need to be performed in order. Each group of positions (0 to 1199 for entry 0, etc.) is connected as the second input to each one of the 64 3-stage cascaded identifiers (the first input being the 1200-input pattern). The circuit does not support dynamic memory updating. Updating of the memory must be completed before the circuit recommences normal operation. This makes the design reported in [2] better in this area since it supports dynamic updating.

As mentioned, we opted for an input pattern that was 1200 bits long, with three stages of cascaded identifiers (bits per symbol were 10, 10, and 12, respectively) and a rule set of 64 rules. The biggest adder used is 4 bits, which can be seen in the operating frequency of the system. The weight of the comparators (value above which the symbol is detected as matched) was set to 6, 6, and 8, respectively, for every level (bits per symbol = 4). This was used to enable the circuit to detect patterns, under the interference of BER = $10^{-1}$. As mentioned, the circuit received the input pattern as an input (along with standard reset and enable signals) and outputs the pattern that matches the closest to an entry in its memory. For its implementation, 205,082 LUTs and 355,013 FFs were used. For only one unit, 3051 LUTs and 5531 FFs (approximately 1/64th) are necessary. The FPGA evaluation board that was used was the ZCU106 Ultrascale+. Since the biggest adder used was 4 bits, an operating frequency of 667 MHz was achieved, providing a total terminal throughput (with no memory updating) of $1200 \times 667 = 800.4$ Gbps. Compared to other pattern-matching circuits, hardware consumption is on the same magnitude but is slightly higher. In a previous study [7], the pattern-matching circuit of the same size of input $\times$ entries consumed 29,056 slices on Virtex 6. This translates to 116,224 LUTs and 232,448 FFs [9], which is about half of that for the proposed design in LUTs and two-thirds of that in FFs. The proposed design, however, provides four times the throughput. In another study [6],

only illustrations are provided for the hardware consumption; however, the throughput provided is still half of the proposed design (for one core). The designs reported in [2–4] are not specialized and the throughput they present is much lower (<30 Gbps). Table 3 presents the aggregated results described above.

**Table 3.** A comparison of terminal throughput (with no memory updating) between the different works.

| Source | Hardware | Terminal Throughput |
|--------|----------|---------------------|
| [7] | 116,224 LUTs/232,448 FFs | 256 Gbps |
| [6] | N/A | 358.4 Gbps [1] |
| [2–4] | Various (not specialized) | <30 Gbps |
| Proposed | 205,082 LUTs/355, 013 FFs | 800.4 Gbps |

[1] The value refers to one processing cluster.

Since the circuit is a kernel implementation, it does not take into account the I/O circuitry. This is left as a choice for the user, and it is heavily dependent upon the type of application that the circuit will realize. The throughput comparison, however, is valid. One previous study [7] uses rules with a much shorter length so that all the I/O pins can be directly mapped on the FPGA, with a very restrained impact on the hardware consumption and throughput. Another study [6] presents only illustrations. The throughput in [2–4] is much lower. In the current case, the board of implementation (ZCU106 Ultrascale+) has multiple I/O devices (multiple I/O pins and gigabit transceivers), which can be used in unison to provide the inputs. The choice of what I/Os are used and how often this happens will determine the end-to-end performance and hardware consumption of the system. Generally, a value of 800.4 Gigabits/s is the highest possible throughput that the synthesizer allows for correct implementation. The design can very easily be run at lower speeds, to provide some leniency for the I/O circuitry. Finally, the output can simply be given as the number of the entry and not the entire pattern, while the input does not necessarily have to be 1200 bits per cycle. With data compression techniques, fewer bits can enter the board, and the larger sequence can be created and fed to the circuit afterward.

## 5. Pattern Matching under Interference and Simulations

### 5.1. Error-Free and Error-Present Input Generation

To test the ability of the system to match patterns under heavy interference, simulations for both error-free and error-present inputs were performed. Since the focus of the design is its ability to detect severely altered patterns with very high throughput, all possible sources of misdetection had to be examined. This includes misfires, in other words, if the design detects a pattern when one is not present. For this reason, to create the input, 100,000 patterns were generated by choosing entries from the memory randomly. For example, the sequence could be {entry 0, entry 8, entry 10, entry 44, entry 21, entry 53, etc.}. Another 100,000 patterns were randomly generated 1200-bit patterns (with a 50% chance of 0 and a 50% chance of 1 for each bit) to make absolutely sure that the design did not detect random patterns as part of the 64 memory entries. For the error-free stream, the memory entries could be anything, but for the error-present stream, additional considerations had to be considered. Firstly, the Hamming distance of each memory entry needed to be sufficiently high for the error rate chosen from all other entries, to ensure that they would not be misidentified. Secondly, the error model had to be robust and flexible, in order to simulate realistic error patterns with a high rate of accuracy (BER = $10^{-1}$).

To ensure sufficient Hamming distance, the 64 entries were constructed with a Hamming distance of 400 bits between each other, using the following method. Assuming that an entry of 1200 bits is made up of 100 sub-patterns of 12 bits of $v_i$, it presents the form $v_1 v_2 v_3 v_4 \ldots . v_{100}$. By performing an exhaustive search on all 12-bit binary numbers it was determined that the biggest set in which all binary numbers have a Hamming distance

of 4 between each other was 128. The memory entries were constructed with this pool of 128 12-bit vectors. Firstly, we aligned the entries vertically.

$$\text{entry 1}: \ v_1^1 v_2^1 v_3^1 \ldots . v_{100}^1$$

$$\text{entry 2}: \ v_1^2 v_2^2 v_3^2 \ldots . v_{100}^2$$

$$\text{entry 64}: \ v_1^{64} v_2^{64} v_3^{64} \ldots . v_{100}^{64}$$

Then, for each column, we shuffled the pool of 128 12-bit vectors and picked 64 at random (no duplicates). These 64 vectors constitute the column $v_i^j, j \in [1, 64]$. All of the vectors have a Hamming distance of 4 between them and, since there are 100 columns, all entries have a distance of 400 between them.

The error model used to simulate the channel is the Gilbert–Eliot two-state Markov chain model [10] which has been shown to be a good approximation of wireless communication channels [11]. Figure 6 presents the model.

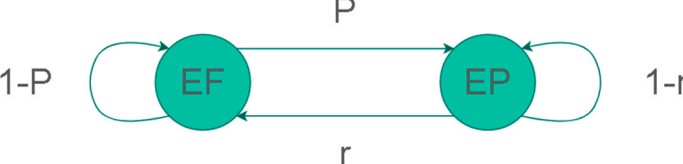

**Figure 6.** Gilbert–Eliot channel model.

The state of EF (error-free) is the state of the channel where no errors occur in transmission, and the state of EP (error-present) is the state where errors occur with a probability of $h$. Transition probabilities between the two states are $P$ and $r$. Both $P$ and $r$ are small while $1 - P$ and $1 - r$ are large. The total percentage of errors is $h \times \frac{P}{P+r} = P_E$. As mentioned, the Hamming distance from one entry to all other entries in the set had to be sufficiently high in order to ensure that the possibility of mistaking one entry for another is inconsequentially low. This model was used to apply a BER of $10^{-1}$ to the generated input.

To summarize the input creation process, first, 64 memory entries with a 400-bit Hamming distance were created. Then, an input stream of 100,000 of these patterns was created by picking any one of them randomly (entry 0, entry 63, entry 4, entry 23, etc.). In all, 100,000 randomly generated patterns were added to the stream, to account for misfires for a total of 200,000 patterns. Without applying the error model, the input stream was used for the error-free case. For the error-present case, multiple values of $P$ and $r$ (Gilbert–Eliot model) were created and, for each value, the error model was applied separately to the error-free input stream, creating as many test input streams as the number of generated values of $P$ and $r$ for the same original error-free input stream. The rate of missed patterns is the average rate of missed patterns of all those input streams.

### 5.2. Error-Free and Error-Present Simulations

For the error-free scenario, no misdetections were reported for all 200,000 patterns, even though the weights of the comparators are not equal to the number of bits per symbol for each level.

For the error-present input, the Gilbert–Eliot Model was applied along the following lines. For a BER of $10^{-1}$, we assumed that $P_E = 0.1$. We substituted $h = 0.5$, as is common practice [10], and we ended up with $r = 4 \times P$. Since $1 - P$ and $1 - r$ were larger than $P$ and $r$, the value of $r$ could not be higher than 0.5. We picked a lower value for the maximum value of $r$, which is 0.4. This gives us a range of 0.1 and lower for $P$. We generate 100 values for $P$ between the range of 0.1 and 0.01, which is the usual range given for the $P$ literature [10–12]. Additionally, the same process was followed for BER $= 5 \times 10^{-2}$. This gives us the values of $r = 9 \times P$ with $P \in [0.044, 0.010]$. The error model was then applied to the previously error-free input stream for each value of $P$ separately (100 simulations) for

both BER values (200 simulations of 200,000 patterns). The average rate of missed patterns was $2 \times 10^{-3}$ for BER = $10^{-1}$ and $10^{-4}$ for BER = $5 \times 10^{-2}$. The biggest losses were reported when the value of P approached 0.1, while the losses were the lowest when the value of P was closer to 0.01. However, the percentage could be improved by increasing the Hamming distance between the memory entries.

### 6. Summary

In this article, we have presented a novel circuit for matching bit patterns in the presence of heavy interference (BER = $10^{-1}$, BER = $10^{-2}$). The circuit accepts one pattern of 1200 bits that suffered errors during transmission as the input and outputs the pattern that was most likely to be transmitted. This consisted of a shift-register memory structure with 64 1200-bit entries and 64 pattern-matching basic-unit circuits. The structure provides throughput in the range of 800 Gbps while consuming a reasonable number of resources, which allows for its implementation on small commercial platforms.

To achieve the maximum operating frequency and low latency, the detection method was implemented through the utilization of a comparator-augmented adder tree-based circuit called the cascaded pattern identifier. Each pattern matching the basic unit received the input pattern of the circuit, along with its respective memory entry. With the use of the cascaded pattern identifier, the process of comparison was achieved with a latency of 23 clock cycles and an operational frequency of 667 MHz. The outputs of all units were then checked, and the detected entry was outputted.

Contemporary methods of pattern matching revolve around the efficient use of memory to provide pattern matching at high speeds while providing the function of updating the memory of the circuit in real time. In its present form, the proposed circuit is the only circuit that can perform pattern matching under interference at such high data rates and that has the potential to be expanded indefinitely without compromising its operational frequency. It also can update its memory, albeit at a much lower pace.

Inherently, apart from bit-pattern matching, the design can also be used without major modifications to detect a series of analog measurements. One example is the case of a group of sensors that produce measurements of a certain bit length. By using comparators and comparing their outputs to a threshold, we can turn each multi-bit measurement into a single bit. This, in turn, transforms the series of events into a bit pattern that can be directly detected by the circuit.

**Author Contributions:** Conceptualization, D.N.; methodology, D.N.; software, D.N.; validation, D.N.; formal analysis, D.N.; investigation, D.N.; resources, D.N.; data curation, D.N.; writing—original draft preparation, D.N.; writing—review and editing, D.N. and P.G.; visualization, D.N. and P.G.; supervision, P.G., C.K. and H.A.; project administration, H.A. and C.K.; All authors have read and agreed to the published version of the manuscript.

**Funding:** This research received no external funding. The APC is covered by a waiver.

**Data Availability Statement:** Not applicable.

**Conflicts of Interest:** The authors declare no conflict of interest.

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
