# Peer review of "High-Throughput Bit-Pattern Matching under Heavy Interference on FPGA"

_electronics, doi:10.3390/electronics12040803_

Round 1

Reviewer 1 Report

A. General idea of the presented scientific work

The paper deals with a formulation of pattern matching algorithm implemented on one of reconfigurable hardware platforms.

The main goal is to reduce the numerical cost (elapsed time) of the algorithm by using specific schemes for comparing the pattern and selected rule (test) sequences of bits.

The presented method is based on the partial parallelization of the pattern comparison operation. Some intermediate coefficients (metrics) are determined to find similarity of the test sequence and elements of the rule sequences. As a result, efficiency of pattern matching scheme is partially improved.

B. Assessment of methodology

I positively evaluate description, explanation of the scheme of the proposed algorithm. The authors presented a discussion of subsequent variants of the scheme, taking into account cascading (layered) structure of the algorithm. Numerical cost of implementation on the FPGA platform (numbers of LookUp Tables and Flip flop components) is discussed. This part is particularly important, interesting for designing a new scheme.

The article lacks the results of tests performed for selected configurations. The results of analyses, numerical tests would be a valuable element of the work. The authors should consider different variants of the patterns in order to determine the limitations of the proposed scheme, critical elements of the proposed method.

C. Form, layout of the paper

The general scheme of the paper is correct. The implemented notation should be modified. There are a lot of errors in notations of variables, coefficients in the description of the scheme.

Detailed remarks regarding errors in notation, form of text and wording are marked in the attached file.

D. Some questions / critical comments on the work

1. Do the authors can determine the critical elements of the presented scheme. Where are some bottlenecks in the configuration, scheme of processing.

2. Do the authors can test the scheme for different starting combination of the input pattern. Does the performance of the scheme depend on the starting point and prepared combination of rules.

3. The article lacks the results of tests performed for selected configurations. The results of analyses, numerical tests would be a valuable element of the work. The authors should consider different variants of the patterns in order to determine the limitations of the proposed scheme, critical elements of the proposed method.

E. Final opinion

The score of implications for research and practice goes to the average. Same major corrections should be implemented. The answer to my comments (three points with some remarks) can improve the quality of the paper and deliver more information for readers.

Author Response

To extensively cover the reviewers comments, a docx files was uploaded.

Reviewer 2 Report

In this paper, a high throughput bit pattern matching under heavy interference on FPGA is presented.

The paper is well-written and contains scientific insights. Only a few suggestions for the authors:

1. To better highlights the improvements deriving from the proposed approach, a comparison with other approaches should be included (summarizing the results in a table form).

2. Possible improvements should be included in the introduction.

Author Response

A docx files has been provided with the answers to the comments

Reviewer 3 Report

The paper presents a FPGA design for matching bit patterns in the presence of interference based on Hamming distance. Long bit sequences are directly compared to many memory entries simultaneously via pipelined adder trees and comparators. The proposed design shows the achieved throughput in the range of 800 Gbps while consuming a reasonable number of resources. The paper is well written and a design idea for under-interference pattern matching circuit is very interesting. However, I have a few questions/comments which I hope the authors address in the revised manuscript.

1. While this work shows very good throughput, as I understand this is still kernel performance, i.e. not including I/O circuitry.  I suggest the authors add end-to-end design, or at least provide some discussions about end-to-end design and performance. Another question is what does the throughput look like when accounting for updating cost of the entire memory structure?

2. I am not clear what would happen if the input pattern has many same closest Hamming distance to different rules in this proposed design.

3. The symbols are stored as entries in the ROM (figure 5). How is the entire pattern is compared to many memory elements in each clock cycle? Please elaborate how the ROM is accessed in your design.

4. Why do the authors use 1200-bit input pattern and a ruleset of 64 rules? How does the circuit perform with longer input pattern and more rules?

5. The authors says "For the error-free stream, 200000 patterns of 1200 bits were produced. 100000 were taken out of the 64 entries randomly and 100000 were randomly generated". This sentence is not very clear to me. Please elaborate.

6. I suggest the authors also put results in tables. It will be easier for readers to follow.

Author Response

Docx files contains the answers to the comments.

Round 2

Reviewer 1 Report

The second version of the paper contains corrections regarding the content, explanations concerning presented scheme of pattern matching implemented on a reconfigurable hardware platform.

The authors introduced some changes. They introduce a lot of suggested modifications, explanations of algorithm and its performance. Still, in my opinion the practical implications of the paper are not clear.

These modifications certainly improve the quality of the paper. I positively assess the changes introduced.  

Unfortunately, in the added text, the authors did not avoid errors and incorrect formulations. It is necessary to improve them (they are marked in the appended file).

The structure of the paper is formally correct: more explanations, some tables with comparison of results.

The authors consistently improve the form of the paper. Now, its layout is correct.

Author Response

We want to thank the reviewer for his positive evaluation and his helpful comments and guidance. All the errors (underlined in brown) in the attacked file have been corrected as instructed.

Reviewer 3 Report

The authors addressed all my comments.

Author Response

We want to thank the reviewer for his/her helpful comments and guidance.